# Effect of Glycerol Concentrations on the Characteristics of Cellulose Films from Cattail (*Typha angustifolia* L.) Flowers

**DOI:** 10.3390/polym15234535

**Published:** 2023-11-25

**Authors:** Nuanchai Khotsaeng, Wilaiwan Simchuer, Thanonchat Imsombut, Prasong Srihanam

**Affiliations:** 1Faculty of Science and Health Technology, Kalasin University, Namon District, Kalasin 46230, Thailand; nuanchai.ko@ksu.ac.th; 2Faculty of Science and Technology, Loei Rajabhat University, Mueang District, Loei 42000, Thailand; wilaiwan.sim@lru.ac.th; 3Department of Rubber and Polymer Technology, Faculty of Science and Technology, Rajabhat Mahasarakham University, Mueang District, Maha Sarakham 44000, Thailand; thanonchat.im@rmu.ac.th; 4Biodegradable Polymers Research Unit, Department of Chemistry, Centre of Excellence for Innovation in Chemistry, Faculty of Science, Mahasarakham University, Maha Sarakham 44150, Thailand

**Keywords:** cattail, cellulose, casting technique, film, hydrophilicity

## Abstract

Plastic waste has become a big problem for the environment globally. Biodegradable polymers are a potential replacement for plastics that can have a positive outcome both environmentally and economically. In this work, we used acid hydrolysis and alkaline treatment to extract cellulose fibers from cattails. The obtained cellulose was used as a substrate for the fabrication of cellulose film using a casting technique on plastic plates. Different concentrations of the plasticizer, glycerol, were used to prepare films for comparison, and its effects on the film’s characteristics were observed. The morphology, chemical structure, and thermal stability of the cattail cellulose (CTC) films were studied using techniques such as scanning electron microscopy (SEM), attenuated total reflection Fourier transform infrared spectroscopy (ATR-FTIR), and thermogravimetric analysis (TGA), respectively. Measurements of transparency, moisture content (MC), water solubility (MS), and water contact angle (WCA) were also performed. Introducing glycerol into the films increased the transparency, MC, and WS values, as well as the gap width between film textures. However, it resulted in a decrease in the WCA of the films, showing that the hydrophilicity of the films is increased by the addition of glycerol. The interaction between the functional groups of cellulose and glycerol was established from the ATR-FTIR and XRD data. The obtained results indicated that glycerol affected the thermal stability and the degree of crystallinity of the produced films. Accordingly, the hydrophilicity of the cellulose film was increased by increasing the glycerol content; therefore, cattail cellulose films can be used as a biodegradable alternative to plastic in the future.

## 1. Introduction

An enormous amount of plastic garbage has been produced by the sharp rise in the production and use of synthetic plastics, causing environmental pollution [1,2]. Plastics have accumulated for decades and are now the world’s biggest environmental hazard due to ineffective trash management [3]. Over time, some plastics are broken down into microplastics, which are then bioaccumulated in the soil or ocean and eventually returned to humans via the food web. Plastics derived from biomass, or “bioplastics”, have attracted interest as a substitute for synthetic plastics. Many reports describe attempts to develop these materials for many applications, including food packaging and biomedical products [4,5]. The main advantages of bioplastics are their renewable and biodegradable properties. Therefore, bioplastics may be environmentally friendly alternatives to traditional plastics and could help to mitigate the plastic pollution crisis [6,7,8]. However, the need for sustainable source of biodegradable materials is one of the guiding factors, as is the development of suitable production methods [9].

Cellulose, which is one of the most abundant biopolymer resources, has been proposed to produce desirable materials as it is a cost-effective and sustainable raw material [10]. The molecular structure of cellulose is made up of glucose units joined by β-1,4-glycosidic bonds. As a biomaterial, cellulose is an environmentally benign polymer that has several advantages, such as low production cost, adaptability of mechanical characteristics, biodegradability, and thermal stability [11,12,13,14,15]. Furthermore, there are a variety of sources from which cellulose can be obtained, including bacteria, plants, algae, and marine life [16,17]. However, the cellulose content varies among the different sources [18]. Previous reports showed that cellulose has been prepared into many different forms, with micro- and nano-crystalline have been extensively used [14,19,20]. The extracted cellulose could be used as a substrate for films [21] and a filler in bio-composite materials [22]. Recently, cellulose has been applied in various fields, including food packaging [23,24], wastewater treatment [25], surface coating materials [26], and biomedicine [27].

Cattail (*Typha angustifolia* L.) is a monocot water weed that grows rapidly in both freshwater and saltwater basins. It typically inhabits wetlands and fresh water [28]. Recently, cattail has spread almost all over the world [29]. The cattail structure contains over 40% fiber, which is composed of 63% cellulose, 8.7% hemicellulose, and 9.6% lignin [30]. However, cellulose content varies in each part of the cattail [31]. The utilization of this fiber in polymer fields might require a new strategy for handling water weeds. To create new cattail-based biomaterials, a deeper comprehension of the characteristics of the components separated from cattail is still required. During the extraction of cellulose, hemicellulose, and lignin are usually removed by alkaline treatment [18]. Oxidation is subsequently performed to bleach the cellulose chains using acid hydrolysis [31]. Cattail is a non-food feedstock that is very productive and has a favorable overall chemical composition of lignocellulose; hence, it has been identified as a promising crop for cellulose bioresources [32].

Generally, biobased polymers have a brittle and fragile texture conferred by their dipole and hydrogen bonding, which form strong intermolecular forces [33]. Previously, many researchers have used plasticizers to improve the toughness, flexibility [34], extensibility, processability [33,35,36], and smoothness of the films [34,37]. Therefore, plasticizers are essential for developing biodegradable polymers with improved properties [38,39,40,41]. Among plasticizers, glycerol is the most frequently used [39,42,43]. Glycerol has several appealing qualities, such as being inexpensive, food-grade, biodegradable, resistant to heat treatment, and non-toxic [44]. Glycerol has also been reported to increase the elongation at break, flexibility, and biodegradability of cellulose-based films [45,46,47]. The structure of cellulose is loosened by the breaking of the intermolecular network when glycerol is added to the films. The insertion of glycerol molecules affects the distances between glucose units in the cellulose. The plasticizer results in lower glass transition temperature of polymers, causing the molecular mobility system to have looser interaction bonds [48]. However, the polymer’s compatibility and polarity [41], as well as the mass content foundation [36,49], determine how effective glycerol is a plasticizer for biopolymers.

The purpose of this work is to use the cellulose that has been separated from cattail flowers by NaOH treatment and bleaching with NaClO before hydrolysis by H_2_SO_4_ to create thin films. The process to extract cellulose in this work could be performed without advanced chemicals and equipment, or hazardous conditions. The obtained cellulose was then cast on plastic plates and left to stand at room temperature for 3 days. To enable comparisons, varying volumes of glycerol were introduced as the plasticizer prior to film casting. Different techniques, such as thermogravimetric analysis (TGA), attenuated total reflection Fourier transform infrared (ATR-FTIR) spectroscopy, and scanning electron microscopy (SEM), were used to characterize the resultant films for further discussion and comparison. The hydrophilicity, transparency, and crystallinity of the films were further examined using X-ray diffraction (XRD) analysis.

## 2. Materials and Methods

### 2.1. Materials

Cattail (*Typha angustifolia* L.) flowers were collected from a pool at Mahasarakham University, Khamriang sub-district, Kantharawichai, Maha Sarakham, Thailand. The cattail plant was chopped, 10 cm from the rhizome and 15 cm from the tip, cleaned of dirt with tap water, and then allowed to dry at room temperature. The dried samples were crushed and stored in plastic bags. Sulfuric acid (H_2_SO_4_), sodium hypochlorite (NaClO), and sodium hydroxide (NaOH) were purchased from Merck Life Science Private Ltd. (Maharashtra, India), LOBA CHEMIE PVT. Ltd. (Maharashtra, India), and Kemaus (New South Wales, Australia), respectively. No additional purification was required for any of the reagent-grade compounds utilized in this investigation prior to usage.

### 2.2. Cellulose Extraction from Cattail Flowers

The cattail blossoms were gathered, submerged in ethanol for an hour, cleaned twice with deionized water, and dried for two hours at 80 °C. After the dried sample was weighed, it was boiled for four hours in a ratio of 1:10 in 4% NaOH (*w*/*v*). Filtration was then performed to stop the reaction. Distilled water was used to wash the obtained solid until a neutral pH was reached, and 2% (*v*/*v*) NaClO was then added for two hours at 80 °C. After centrifuging the reaction mixture to extract the residue, 5% H_2_SO_4_ was added and hydrolyzed for three hours at 50 °C. After filtering and neutralizing the pH, the extracted cellulose was stored in a refrigerator in preparation for further processing.

### 2.3. Preparation of Cellulose Films from Cattail Flowers 

Ten milliliters (mL) of distilled water were mixed with 2.5 g of the obtained cellulose. After that, the mixture was agitated for an hour at room temperature to achieve homogeneity. A polystyrene petri dish, 9 cm in diameter, was filled with the homogenous cellulose suspension, which was then allowed to dry for three days at room temperature. Four distinct films containing 0.14, 1.38, 2.76, and 4.14% (*v*/*v*) of glycerol were made by first mixing the glycerol and extracted cellulose, stirring until homogeneous, and pouring on the plates. The plates were then dried in the air for three days, as for the preparation of native film. After removing the prepared cattail cellulose (CTC) film from the plastic plate, it was stored in a desiccator until further examination. Figure 1 provides a summary of the processing utilized in this work.

### 2.4. Analysis of Film Characteristics

#### 2.4.1. Transparency of Films 

A UV-Vis spectrophotometer (Lambda 25, Perkin Elmer, MA, USA) was used for the determination of the transparency of the prepared CTC films, as previously described [50]. In short, the films were cut into rectangles and placed directly into the spectrophotometer cell. To obtain the average film transparency, the percentage transmittance of light at 660 nm through each film was measured three times.

#### 2.4.2. Morphology Observation

The morphology of the produced CTC films was studied using a scanning electron microscope (SEM) (JEOL, JSM-6460LV, Tokyo, Japan). A cross-section of film was cryogenically frozen by immersion in liquid nitrogen. Each dried film was adhered to an aluminum stub. The film surfaces were sputter-coated with Au to excite electrons and analyzed using a SEM at 15 kV.

#### 2.4.3. Functional Group Analysis

To analyze the functional groups of the CTC films, a Fourier transform infrared (FTIR) spectrometer (Invenio-S, Bruker, Karlsruhe, Germany) equipped with an attenuated total reflectance (ATR) accessory was applied. Each ATR-FTIR spectrum was obtained using 32 scans, with air serving as the reference, across the wavenumber range of 4000–400 cm^−1^ at a spectral resolution of 4 cm^−1^.

#### 2.4.4. X-ray Diffraction Analysis 

The CTC films prepared with different glycerol concentrations were examined using X-ray diffraction (XRD, Bruker D8, Karlsruhe, Germany). The diffraction angle was changed from 2θ = 5° to 60°, at a step size of 0.02°/s, with Cu Kα, λ = 1.5406 Å, 40 kV, and 40 mA. The previously published Formula (1) was used to determine the crystallinity index (CI) [51,52].
CI= [I200 − Iam/I200] × 100 (1)
where Iam is the intensity of the amorphous peak around 2θ = 18° and the 22.5° (2 0 0) peak, and I200 is the intensity of the crystalline peak.

#### 2.4.5. Thermal Behavior Determination 

The CTC films’ thermal behavior was examined using a thermogravimetric analyzer (TGA) (SDTQ600, TA-Instrument Co. Ltd., New Castle, DE, USA). Three to five milligrams of film were heated in the range of 50 to 800 °C at a fixed rate of 20 °C per minute in a nitrogen environment. The weight decreases were noted at several points in time.

#### 2.4.6. Hydrophilic Properties of CTC Films 

Measurement of the water solubility (WS) and moisture content (MC) revealed the hydrophilic properties. By monitoring the weight loss of the films as previously mentioned, the MC of the CTC films was ascertained gravimetrically [53]. CTC films (2 × 2 cm^2^) were weighed both before (W_1_) and after drying for 24 h at 80 °C in an oven (W_2_). The average moisture content of each film was measured on three duplicates. The following Equation (2) was used to determine each film’s MC (%).
MC (%) = [(W_1_ − W_2_)/W_1_] × 100(2)

WS was evaluated by cutting the film into rectangular shapes (2 × 3 cm^2^) and drying at 100 °C until the observed weight remained constant (W_1_). After the test films were added to a test tube with 10 milliliters of distilled water, the mixture was stirred and allowed to sit at room temperature for a full day. The undissolved films were dried absolutely at room temperature after immersion, and the undissolved films (W_2_) were weighed. For every duration, the measurements were performed three times. The water susceptibility (%) values were determined using Equation (3) as follows:WS (%) = [(W_1_ − W_2_)/W_1_] × 100 (3)

In addition, utilizing a water contact angle (WCA) analyzer (model OCA 11, DataPhysics Instruments GmbH, Filderstadt, Germany), the hydrophilicity of the CTC film surfaces was determined from the WCA of the films. The films were divided into rectangles of 3 × 5 cm^2^ and set up on a horizontal moving platform that had been coated in black Teflon and was equipped with a WCA analyzer. Using a microsyringe, an appropriately sized droplet of water (5–10 μL) was applied to the film’s surface. The water droplet’s contact angle was measured. For the results, averages of the triplicate measurements for each sample were determined.

## 3. Results and Discussion

### 3.1. Appearance of CTC Films

Following chemical treatments of boiling with NaOH, bleaching with NaClO, and acid hydrolysis, 18.37 ± 0.86% of the cellulose was recovered from the cattail flowers. The obtained cellulose content was in agreement with previous reports [18,31]. The flowers contain practically pure cellulose, even if the cellulose content is lower than that of other cattail parts. One of the relevant film properties to be considered for their practical use is transparency and water solubility, especially in the food packaging industry [9,50,54]. This macroscopic aspect of the CTC films was visually evaluated by covering letters with each prepared film, with and without glycerol, as shown in Figure 1. Overall, both films are white in color, slightly shiny, and highly transparent. There was no discernible change in transparency between the CTC films with (Figure 1A) and without glycerol (Figure 1B–D). However, the film color slightly changed from cloudy white to light yellow. The texture of the glycerol-plasticized film was homogeneous, suggesting that glycerol was well-dispersed and helped to blend the cellulose molecules together. The films were successfully fabricated, confirming that the process used in this work could be used practically to extract cellulose for film preparation. This result agrees with other reports [31,55,56,57]. Moreover, the CTC films were tested for light transmittance (T666) and high values were obtained, as shown in Table 1. It can be noted that when the glycerol concentration increased, values of film transparency, MS, and WS gradually increased. However, the WCA of the films gradually decreased. The hydroxyl groups of glycerol, which reacted favorably with water and increased the hydrophilicity of the film surfaces, would be the reason for this outcome [54]. The CTC film with the highest glycerol content had a high WS percentage and retained about 80% of its original weight after 7 days. The outcomes demonstrated how the cellulose film’s poor water solubility was impacted by its strong structural integrity [58,59,60]. In contrast, the cattail cellulose films with and without glycerol had a low moisture content, of not over 5% (Table 1). This shows that the cattail film is lightweight and might be a good material for water resistance. It may be possible to use this film as food packaging material, especially for vegetables and fruits.

### 3.2. Morphological Determination

SEM micrographs of the CTC films with and without glycerol are shown in Figure 2. The SEM scale bars show that the isolated cellulose fibers were arranged in microcrystalline strands measuring three to six micrometers. Even at high magnification, the native CTC film’s cross-section (Figure 2aI) revealed a smooth densely packed texture devoid of phase separation. Smooth, flat, and dispersive fibers were located on the side attached to the petri dish (Figure 2aIII), whereas flat incomplete fibers imbedded into the film were visible on the top surface of the native film (Figure 2aII). Compared to the original film, the cross-section of CTC films plasticized with glycerol (Figure 2bI–2dI) had a looser texture. When glycerol content increased, a wider gap between film textures was observed. This indicated that glycerol, as a small hydrophilic molecule, was well-integrated between cellulose chains and retained water molecules that were removed in the drying process, resulting in an increase in the free volume in the films [61]. In addition, chemical bonds formed among glucose units and glycerol should affect the homogeneous texture of the films [62]. However, when observed on the top surfaces (Figure 2aII–2dII) and the side attached to Petri dishes (Figure 2aIII–2dIII), the glycerol-containing films did not significantly differ from the films without glycerol. The short and tiny strands were deeply ingrained in the texture of the film.

### 3.3. Functional Group Analysis

The functional groups and molecular interactions between the constituents in the produced films were examined using ATR-FTIR spectroscopy. Figure 3 shows the results of the ATR-FTIR analysis that was conducted on the cattail cellulose films. The top spectrum (Figure 3(a)) showed characteristic bands of the functional groups of cellulose for -OH stretching, C=O stretching, C-H stretching, and C-O-C (β-(1-4)-glycosidic bond) at 3333, 1727, 1315, 1028, and 896 cm^−1^, respectively [18,48,63]. Furthermore, C-H stretching is indicated by the absorption bands at 2914 and 2850 cm^−1^ [49]. The spectra of the cellulose films plasticized with glycerol (Figure 3(b–d)) have some notable differences from the native cellulose film. The characteristic bands between 800–1150 cm^−1^ are slightly sharper when increasing the glycerol content. This means that introducing glycerol into the cellulose might cause an interaction between the functional groups of the cellulose and the -OH groups of the added glycerol [60,64,65]. The absorption bands of -OH stretching and C–H stretching were also observed at 3333 cm^−1^ and 2914 and 2850 cm^−1^, respectively. These characteristic bands progressively intensified as the glycerol level increased. The remarkable absorption bands at 925 cm^−1^ were found only in the glycerol-plasticized films. The insertion of glycerol molecules in the film matrix may cause an increase in hydroxyl groups, as indicated by the increases in the -OH stretching signal.

### 3.4. Determination of Thermal Properties

A thermogravimetric analyzer (TGA) was used to investigate the thermal stability of the CTC films. According to the TG thermograms, weight loss occurred in at least four stages (Figure 4). The first weight loss was caused by the film losing moisture, which happened at low temperatures (60–100 °C) [66]. As a result of glycerol evaporation, the second stage of weight decreased from 120 to 280 °C. For the film with the highest glycerol concentration (4.14%), the weight loss recorded at this stage was roughly 25%, with no substantial loss of the native cellulose film observed until the next step. The third stage (280 to 400 °C) was related to cellulose degradation [46]. Finally, the degradation of other lignocellulosic fibers occurred in the fourth stage (400–500 °C), as indicated by small peaks. At the end temperature (800 °C), the charred residue content of native film had the highest value, of about 20%, followed by the cellulose films mixed with glycerol at 1.38%, 2.78%, and 4.14% (*v*/*v*), which contained a charred residue content of 12%, 10%, and 8%, respectively. This remaining substance was carbon charcoal that did not degrade further. Additionally, it is evident that the native cellulose film had a higher initial degradation temperature than the films that had been plasticized with glycerol. As the glycerol content increased, the glycerol-mixed films exhibited greater thermal stability when considering the maximum decomposition temperature (T*_d, max_*). The DTG curves in Figure 5 illustrates that the films with 2.78% and 4.14% glycerol exhibited T*_d, max_* at approximately 355 °C, whereas the temperature in the native film was 337 °C. Some parameters on the thermal behavior of all films are summarized in Table 2. Because robust intermolecular interactions were established between cellulose and glycerol, the degradation temperature of the glycerol-plasticized films increased, improving their thermal stability [9,18,46,67]. In addition, the T*_d, max_* at 209 and 213 °C was the decomposition temperature of glycerol, which was prominent in the glycerol-plasticized films, especially those containing 2.78% and 4.14% of glycerol.

### 3.5. Crystallinity of Films

Using X-ray diffraction, the crystallinity of the CTC films with and without glycerol was examined. The XRD pattern of the native CTC film showed diffraction peaks at around 14.5°, 16.5°, and 22.5°, which matched the planes of (110), (1–10), and (200), respectively. These correspond to the crystallographic planes of the cellulose I pattern, as shown in Figure 6(a) [68]. The addition of glycerol significantly reduced the intensity of the diffraction peak at 22.5°, while the peak at 12.3° became more prominent. The presence of glycerol also affected the 14.5° peak: the intensity increased when glycerol was added at 1.38% (Figure 6(b)) and 2.78% (Figure 6(c)) but dropped considerably at 4.14% of glycerol (Figure 6(d)). Overall, the CTC films’ crystallinity declined as the glycerol content rose. As determined from Equation (1), the crystallinity values of the CTC films were 84.65%, 84.15%, 82.12%, and 74.52% for the CTC films with 0%, 1.38%, 2.78%, and 4.14% glycerol, respectively. This indicated that glycerol interfered with cellulose polymeric chain packing and resulted in less-ordered molecular organization. However, the crystallinity of cattail cellulose extracted by this work was in line with the crystallinity of cellulose nanocrystals in previously reports [69].

## 4. Conclusions

In conclusion, it is possible to successfully extract the cellulose fibers from cattail flowers and use them to create biodegradable films that are transparent and have good light transmission properties. After the cellulose material was extracted using a standard chemical treatment technique, it could be further processed into films. The outcomes demonstrated the validity, ease of use, and economy of the current procedure. Compared to the original cellulose film, the films made with glycerol added as the plasticizer were less densely packed. This was reflected by the decreased degree of crystallinity. The interactions between the functional groups of the cellulose and the hydroxyl groups of the glycerol were demonstrated by ATR-FTIR spectroscopy. The results from SEM micrographs indicated that glycerol affects the texture of the CTC films. These effects were associated with the increased thermal stability and hydrophilicity of the films. The prepared films partially disintegrated in 7 days, indicating a certain degree of structural stability with potential degradability in the environment. The presence of glycerol helped increase the hydrophilicity of the films, which affects the water content and solubility of the films. Although further studies are still needed, our study shows that cattail cellulose is a promising material that could serve as a renewable and biodegradable alternative to reduce the use of synthetic plastics.

## Data Availability

Data are contained within the article.

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
