# Peer review of "Effect of Glycerol Concentrations on the Characteristics of Cellulose Films from Cattail (Typha angustifolia L.) Flowers"

_polymers, 2023, doi:10.3390/polym15234535_

Round 1

Reviewer 1 Report

Comments and Suggestions for Authors

The article ‘Effect of Glycerol Concentrations on the Properties of the Films 2 Cellulose from Cattail (Typha angustifolia L.) Flowers’ is an interesting piece of work. Although, the objective of the article is somewhat clear; however, the usefulness of the developed material is unclear. In addition the novelty is not clear.

Hence, the article in the present form cannot be accepted in the present format.

Some of the major comments and suggestions are as follows:

1.   1. In ‘Introduction’ section of the article, the authors need to explain in more acceptable way regarding why they have chosen Cattail (Typha angustifolia L.) as a source of cellulose ? What are the advantages for using this water weed than the other sources?

2.     2. It is not clear that, why glycerol needs to be used as plasticizer??  Author mentioned that ‘Glycerol composed of various at- 70 tractive properties including non-expensive, biodegradable, resistant to heat treatment, 71 non-toxic and food grade’ . Will author utilized glycerol for introducing ‘biodegrability’??

3.     3.  There have been lots of works already pubiished regarding sustainable production and extraction of cellulose from different plant sources. Hence, authors need to clarify why the mentioned process is advantageous than the other.

4.     4. The figures inside the scheme.1 are not clear.

5.  5. In the ‘Material and Method’ section, under 2.3 Preparation of Cattail Cellulose Films, the author mentioned ‘The prepared cattail cellulose film 109 (CTC) were peeled off the plastic dishes’, there will be high chance to damage the internal structure and edges of the films. How authors ensured there was no damage to the films???

6.  6. For SEM, authors need to mention how they have prepare samples for crosssectional morphological studies.

7.     7.  Figure.6 needs to be changed.

8.  8. Conclusion section is ambiguous. Authors need to focus what advantages this process brings to the existing methodologies.

Comments on the Quality of English Language

Moderate English language editing is required. 

Author Response

Dear reviewer,

Thank you for your comments and suggestions. Please see the attached file of responses.

Reviewer 2 Report

Comments and Suggestions for Authors

The manuscript polymers-2702456 presents the preparation and characterization of films based on cellulose isolated from cattail flowers, improved with glycerol at different concentrations, by using casting technique.

The manuscript still needs a lot of improvements, and some of the problems I noticed are listed below:

Abstract

- Please clearly explain how the films were obtained! Nothing is mentioned about acid hydrolysis!

- The Abstract section must be improved with clearer explanations!

Keywords

- “Characterization” is a very general Keyword! Please revise it!

Introduction

- L.62: Reference [18] is written differently! Please correct it!

- L. 72-74: This sentence is ambiguous! Please revise it!

- L. 79: Please exactly mention “these chemical treatments”!

- Please explain clearly how the authors obtained the films, so that the readers can understand the too!

- Please add in Introduction section also data related this subject, films based on cellulose and glycerol! The authors cannot discuss only general information, without emphasizing what they bring new compared to the data from literature!

2. Materials and Methods

2.3. Preparation of the Cattail Cellulose Films

- L. 108-109: There is no clear explanation related to the preparation of the films with glycerol! Please explain how was prepared these films!

2.4. Transparency of Films

- L. 114-119: This method must be moved to the section 2.5. “Characterization of the Films”, because is not related to the preparation of the films!

2.5. Characterization of the Films

2.5.1. Morphology observation.

- L. 122: “The prepared CTC films observed their morphology”? Please revise it!

2.5.2. Functional Group Analysis

- L. 128: “All FTIR spectra were recorded” must be “All ATR-FTIR spectra were recorded”!

2.5.3. X-ray Diffraction Analysis.

- L. 131: Please remove the repetition for “were characterized”!

- L. 132-133: “Xray diffraction (XRD, Bruker D8 advance with Cu Kα, λ = 1.5406 Å, 40 kV 132 and 40 mA), Germany”? There is a mixture between the name of the equipment and parameters! Please make the adequate correction!

- L.135-137: The authors present the same formula for establishing the crystallinity degree in two ways that only contain different notations. Please explain clearly how the crystallinity degree was determined by using a single formula!

- Please review the entire XRD section!

2.5.5. Hydrophilic Properties of CTC Films.

- L. 150-156: Did the authors use magnetic agitation when tested the WS?

3. Results and Discussion

3.1. Appearance of CTC Films

- L. 166: The same observation for “chemical treatments”! There is no information related to the characteristics of the obtained cellulose! Please briefly explain the entire process for cellulose preparation!

- There is no information related to the obtained cellulose! Did the authors obtain microcrystalline cellulose or nanocrystalline cellulose? Please add a separate section to characterize the cellulose obtained! Information about the material obtained after each mentioned chemical treatment must be added!

- L. 171-173: “CTC films was visually evaluated”? Please add in Figure 1 pictures for all samples!

- L. 177-178: “The films were successfully fabricated, confirming that the chemical treatment could be used practically to extract cellulose for film preparation”?? What chemical treatment?

- L. 182: “the films transparency slightly decreased while those of MS, WS and WCA were gradually increased”? The transparency of MS, WS and WCA? Please revise this sentence and discuss separately each characteristic!

- L. 186-187: “had good structural integrity affected on low water solubility”? Please revise it!

- How can the authors explain the fact that with an increase of the glycerol content within films, the moisture content of the samples increases at the same time as the water contact angle (contact angle is higher than 90°)?

- Please add a new figure where to show the droplet contact angle shapes for all films!

3.2. Morphological Determination

- L. 198: “Scanning electron micrographs of the CTC films”! If the authors already mentioned the abbreviation SEM, please use it in the entire manuscript!

- L. 204-205/L. 208-209: There is a contradiction between “More cracks and pores were observed as the content of glycerol increased” and “chemical bonds formed among glucose units and glycerol would be concerned the homogeneous texture of the films”! Please clarify this assumption!

- Why did the authors do SEM micrographs for "top surfaces and the side attaching to Petri dishes"? What notable differences did the authors want to observe between these two figures? From my point of view, authors should only keep SEMs for "top surfaces"!

- L. 214-215: Please add “%” for all numbers at caption of Figure 2! The same observation for all the figures in the manuscript!

3.3. Functional Groups Analysis

- L. 218: The same observation for “Attenuated reflection-Fourier transform infrared (ATR-FTIR) spectroscopy”!! If the authors established an abbreviation, this must be used in the entire manuscript! Please revise carefully the entire manuscript and use only the abbreviation, right after the first mention!

- L. 221-222: “The top spectrum (Figure 3a) showed functional group peaks for”?? “functional group peaks”? The authors must indicate the sample’ name when discuss the characteristic bands, in this case cellulose! Moreover, the authors must discuss about "bands" and not "peaks"!

- L. 233-234: Please improve Figure 3! The characteristic bands are quite difficult to identify, in the form in which they are presented now! Please make ATR-FTIR spectra wider and clearer!

- L. 226-227: “The peaks at between 800 and 1150 cm−1 are assigned to the glycerol”?? Have the authors heard about the "fingerprint" region in FTIR spectra of cellulose? Please revise this assumption!

- L. 229: “outstanding peaks at 925 cm−1”? What do the authors mean by "remarkable"? This word is not used properly!

- The authors must add the FTIR spectrum of glycerol, so that it can be compared with that of the samples!

- Moreover, in FTIR spectroscopy, there are characteristic absorption bands and not peaks! Please make the adequate correction to the entire section!

- This section must be rigorously reviewed and correct assumptions must be made!

3.4. Thermal Stability

- L. 246: “the remaining weight of native film”? Maybe the authors refer to the charred residue?

- L. 251: “the maximum temperature of decomposition rate (Td, max)”?? Decomposition temperature or decomposition rate? There is a mix between the terms, which makes me conclude that the authors do not know the basic notions of the thermal degradation of samples!

- Please add a table where all the parameters (onset degradation temperature, maximum temperature of decomposition, etc) must be mentioned in order to be easier to compare!

3.5. Crystallinity of Films

- L. 269: “14.5°, 16.5° and 22.5”! These correspond to the crystallographic planes of cellulose, (110), (1-10) and (200)! Please add also this information, if these are used in Figure 6!

- L. 281: In Figure 6 there is a mixture between different notations of the crystallographic planes of cellulose, such as: the new notations, (110), (1-10) and (200) and the old notations, (101), (10-1), and (002)! Please use one of these notations!

- L. 277-278: “This indicated that glycerol interfered with cellulose polymeric chain packing and resulted in less ordered molecular organization”?? How did the authors reach this conclusion, if in the ATR-FTIR method, the authors mentioned that more hydrogen bonds were established between the cellulose and glycerol (L.230-232: “hydroxyl groups in glycerol formed strong interactions with the hydroxyl groups in cellulose, via the intermolecular hydrogen bonding”), which would lead to an increase in crystallinity.

4. Conclusions

- L. 288-289: “This was expressed by decreasing of crystallinity percentages”?? Maybe crystallinity degree?

- The Conclusions section must be improved with accurate data from the manuscript!

In my opinion, this manuscript presents serious difficulties in the interpretation of data and some of the foundations are unsafe! In conclusion, the manuscript must be rigorously checked before to be recommend for publication in Polymers journal!

Comments on the Quality of English Language

 Moderate editing of English language required

Author Response

Dear reviewer,

Thank you for your comments and suggestions. Please see the attached file for a response.

Reviewer 3 Report

Comments and Suggestions for Authors

The experimental article “Effect of Glycerol Concentrations on the Properties of the Films Cellulose from Cattail (Typha angustifolia L.) Flowers” is devoted to the current direction - the search for alternative packaging to plastic. The article corresponds to the profile of the Polymers publication, but has a number of shortcomings, after which the article can be published.

Notes:

1. Line 58: The sentence needs to be rephrased, as it is, it is ambiguous.

2. It would be nice to provide a photo of the raw material - Cattail. It would be appropriate to present a photograph in materials and methods (section 2.1).

3. The authors should have given the component composition of the feedstock before the isolation of cellulose, and then the processed cellulose.

4. Despite the fact that the authors provided an excellent diagram 1, it is necessary to provide an image of 4 films, which will facilitate understanding of the article and decorate the article.

5. Line 167: it is not clear what the authors meant about the phrase: “data not given”?

6. Lines 167-169 are not relevant in the discussion, since this is not the result of this study, but an analysis of the literature. Is it necessary to move this text into the introduction and explain to the reader why Cattail flowers were chosen in this study if their cellulose content is lower than in trees?

7. The most significant remark regarding this article is the lack of strength characteristics of the films obtained. They are supposed to be very fragile.

Author Response

(The authors gave the same response as above.)

Round 2

Reviewer 2 Report

Comments and Suggestions for Authors

The manuscript polymers-2702456 has been improved over the previous version, but still have some corrections must be done:

- L. 85-89: “The structure of cellulose, which is composed of high hydroxyl groups, can form important bonds like hydrogen bonds and break the intermolecular network in the polymers, resulting in an increase in the molecular mobility system but a decrease in the glass transition temperatures [48]. This action might introduce the biodegradability of cellulose.”? Even if I mentioned that the sentence is ambiguous, the corrected form of the sentence is even more ambiguous!

- What is “high hydroxyl groups”?

- Who can “break the intermolecular network in the polymers”? The authors refer to the hydrogen bonds? Please clarify this assumption!

- What “action might introduce the biodegradability of cellulose”? This sentence must be removed!

- The reference [48]: “Poly(Lactic Acid)/Coplasticized Thermoplastic Starch Blend: Effect of Plasticizer Migration on Rheological and Mechanical Properties” it is not related to cellulose!! It doesn't even contain the word cellulose! Please add a valid reference!

- Please make the adequate corrections to this sentence!

- L. 98: “Attenuated total reflection Fourier transform infrared (ATR-FTIR) spectroscopy” and not “attenuated reflection FTIR”!

- L. 167: “Iam is the intensity of the amorphous peak” and not “Iam is the intensity of the amorphous valley”!

- L. 168: “I200 is the intensity at the diffractogram's maximum” must be either “I002 is the maximum intensity of the (002) lattice diffraction” or “intensity of the crystalline peak”!

- L. 205: “The obtained” and not “The obatined”!

- L. 406: “ATR-FTIR spectra” and not “ATR-IR spectra”!

- L. 404-406: “change the absorption bands of the film”?? In my opinion, it would be better to mention only: “The interaction between the functional groups of the cellulose and the -OH groups of the glycerol, was demonstrate by ATR-FTIR spectroscopy”!

- L. 406-407: ATR-FTIR spectra do not give information about “cellulose texture”! Please revise the sentence!

- L. 407-408: “These relate to the thermal stability and hydrophilicity of films.”?? This sentence isn’t clear! Please revise it!

Comments on the Quality of English Language

Moderate editing of English language required

Author Response

We have revised and improved following all your comments and suggestions from reviewers. The detailing changes for this round are highlighted in yellow and summarized as follows:

All references are relevant to the contents of the
manuscripts have been checked.

  • Lines 85–90 have been revised.
  • Lines 99–100 have been revised.
  • Lines 169–170 have been corrected.
  • Line 207 has been corrected.
  • Lines 415–418 have been revised.

Reviewer 3 Report

Comments and Suggestions for Authors

The authors took my comments into account. Although the ideology of the experiment is ambiguous, the work is of interest and can be published.

Author Response

We appreciate your comments and suggestions. We would like to construct our next experiment with a clear ideology and goals.
